# Clinical Characteristics of Patients with SARS-CoV-2 N501Y Variants in General Practitioner Clinic in Japan

**DOI:** 10.3390/jcm10245865

**Published:** 2021-12-14

**Authors:** Mariko Hanafusa, Jin Kuramochi, Katsutoshi Ishihara, Makiko Honda, Nobutoshi Nawa, Takeo Fujiwara

**Affiliations:** 1Department of Global Health Promotion, Tokyo Medical and Dental University, Bunkyo-ku, Tokyo 113-8510, Japan; tkmrthsr@tmd.ac.jp; 2Kuramochi Clinic Interpark, Utsunomiya City, Tochigi 321-0114, Japan; kuramochi59naika@gmail.com; 3Department of Radiology, Dokkyo Medical University, Shimotsuga Gun, Tochigi 321-0293, Japan; seikenduki@gmail.com; 4Department of Radiology, The Fraternity Memorial Hospital, Sumida-ku, Tokyo 130-8587, Japan; hnddrnm@tmd.ac.jp; 5Department of Medical Education Research and Development, Tokyo Medical and Dental University, Bunkyo-ku, Tokyo 113-8519, Japan; nawa.ioe@tmd.ac.jp

**Keywords:** severe acute respiratory syndrome coronavirus-2, COVID-19, mutation, N501Y, general practice

## Abstract

The clinical characteristics of patients with N501Y mutation in SARS-CoV-2 variants (N501YV) is not fully understood, especially in the setting of general practice. In this retrospective cohort study, COVID-19 patients admitted to one general practitioner clinic between 26 March and 26 May 2021 were retrospectively analyzed. The characteristics, clinical symptoms and radiological findings before treatment were compared between N501YV and wild-type 501N. Twenty-eight patients were classified as wild-type 501N and 24 as N501YV. The mean (±standard deviation) age was 37.4 (±16.1) years, with no significant difference between groups. Among clinical symptoms, prevalence of fever of 38 degrees Celsius (°C) or higher was significantly higher in the N501YV group than in the wild-type 501N group (*p* = 0.001). Multivariate analysis showed that fever of 38 °C or higher remained significantly associated with N501YV (adjust odds ratio [aOR]: 6.07, 95% confidence interval [CI]: 1.68 to 21.94). For radiological findings, the lung involvement area was significantly larger in patients infected with N501YV (*p* = 0.013). In conclusion, in the N501YV group, fever of 38 °C or higher and extensive pneumonia were more frequently observed compared to the wild-type 501N group. There was no significant difference in terms of other demographics and clinical symptoms.

## 1. Introduction

The coronavirus disease 2019 (COVID-19), caused by the severe acute respiratory syndrome coronavirus-2 (SARS-CoV-2), has induced high number of deaths worldwide [1]. The emergence of viruses that have acquired genomic mutations makes the SARS-CoV-2 pandemic even more difficult to control [2]. Around the 9th week of 2021, Japan entered the fourth wave of the SARS-CoV-2 pandemic with a rapid replace of predominant strains with the B.1.1.7 lineage of SARS-CoV-2 with a N501Y replacement in the receptor binding domain (RBD) of spike (N501YV). The number of new cases per week peaked at approximately 32 per 100,000 population at 19th week of 2021 [3]. The national whole genetic screening of COVID-19 cases at that time showed that the proportion of the B.1.1.7 lineage had reached over 80% [3].

The B1.1.7 lineage was presumed to be structurally more transmissible [4,5], and clinically, an increase in reproduction numbers was reported. Previous study in the UK, which estimated the transmissibility of the B1.1.7 linage using a variety of statistical and dynamic modeling approaches, reported that the reproduction number was 43 to 90% (95% confidence interval [CI]: 38 to 130%) higher than that of the preexisting variants [6]. In addition, there is evidence of the severe impact of the B1.1.7 lineage. In the UK, S gene target failure (SGTF) was monitored as a proxy for the B1.1.7 lineage [7], and in a community-level matched cohort study, SGTF had a mortality hazard ratio of 1.64 (95% CI: 1.32 to 2.04) compared to the previously circulating variants [7]. In an analysis of the data on COVID-19 cases reported to The European Surveillance System (TESSy), the authors found higher odds of the B.1.1.7 lineage/SGTF compared to the non-variants of concern (VOC) cases in intensive care unit (ICU) admission (aOR: 2.3, 95%CI: 1.4 to 3.5) [8]. There is a realistic possibility of the impact of infection with the B.1.1.7 lineage on severity compared to infection with non-VOC [9].

However, detailed clinical characteristics such as symptoms presented to a general practitioner before intensive treatment have not been investigated in previous studies, despite their value for the decision-making process of healthcare policy or caring management. When it comes to the clinical characteristics of VOC cases, there is a single-center report from Marseille, France; 211 patients in conventional infectious disease units infected with N501YV showed higher rates of fever (aOR: 2.58, 95%CI: 1.72 to 3.86), and lower rates of rhinitis (aOR: 0.50, 95%CI: 0.33 to 0.76) and anosmia (aOR: 0.57, 95%CI: 0.36 to 0.90) compared to those infected with non-VOC [10]. In a report from England’s Office for National Statistics, interviews conducted on 6000 randomly selected people who had been positive for SARS-CoV-2 PCR test in community level found that cough, sore throat, fatigue, myalgia and fever showed the largest differences rates in reported symptoms during the COVID-19 incidence between SGTF and non-SGTF [11]. However, previous references did not provide sufficient clinical characteristics of N501YV, as only the main symptoms at the time of observation were investigated, or the collected data were past symptoms the interviewees recalled. A report from the Chinese Center for Disease Control and Prevention showed that 81% of COVID-19 cases were classified as mild disease [12], and managed at home or in designated non-health facilities [13]. This makes it difficult to study the clinical characteristics of this patient population. In Japan, SARS-CoV-2 is classified as a “designated infectious disease”, which allows the Government to legally recommend hospital admission for patients with COVID-19 regardless of disease severity. Therefore, in Japan, it was able to conduct detailed observation of patients with non-severe to critical illnesses who are not hospitalized in other countries.

The purpose of this study was to compare the differences in clinical characteristics and radiological findings of admitted COVID-19 patients, before treatment at a general practitioner in Japan, between N501YV and wild-type 501N cases.

## 2. Materials and Methods

### 2.1. Data Source and Study Design

This study was conducted at the Kuramochi Clinic Interpark, in Utsunomiya City, Tochigi, Japan. The sample size was calculated based on the expected prevalence of fever over 38 °C in patients infected with wild-type 501N (25% to 40%) in non-critical COVID-19 population, expecting an odds ratio to be 5.0 to 8.0 for patients infected with N501Y compared with wild-type 501N. Using a two-tailed test with a significance level of 0.05 and a statistical power of 80%, a maximum sample size of 54 was obtained. Expecting 10% dropout, a sample size of 60 or more was estimated to be sufficient. A total of 64 patients who were detected SARS-CoV-2 by real-time reverse transcription-PCR (RT-PCR) test with a Ct value cutoff of 40 of saliva or nasopharyngeal swabs, in accordance with the Pathogen Detection Manual of the National Institute of Infectious Diseases (Japan) [14], were consecutively hospitalized from 26 March to 26 May 2021. Sixty patients, excluding each 2 patients who had not consented to participate in the study or who did not have appropriate test samples, were screened for N501YV by a specific real-time RT-PCR in our institute (*n* = 57) or in the Public Health Institute (*n* = 3). Cases that were undetectable by both wild-type 501N and N501YV detection systems were excluded (*n* = 8).

Demographic and clinical data were retrospectively collected from medical records. Demographic information included age, sex, body mass index (BMI), smoking status, high-risk medical condition for severe COVID-19 (i.e., type 2 diabetes mellitus, hypertension, cardiovascular disease, chronic obstructive pulmonary disease, chronic kidney disease, cancer, obesity and dyslipidemia [15,16,17,18]) and birth country. Based on previous literature, BMI was categorized into four groups: underweight was defined as a BMI < 18.5 kg/m^2^, normal weight as 18.5–23.9 kg/m^2^, overweight as 24.0–27.9 kg/m^2^, and obesity as ≥28 kg/m^2^ [18]. Missing data for BMI and smoking status were treated as dummy variables. Clinical data included daily subjective symptoms, vitals, radiological findings on CT scan, laboratory results, transmission route, pre-hospital management, outcome and the date of onset, COVID-19 diagnostic testing, admission and treatment started. According to the National Institutes of Health (NIH) guidelines, the patients were categorized into five based on clinical grade of disease severity (i.e., asymptomatic or pre-symptomatic infection, mild illness, moderate illness, severe illness and critical illness) [19].

Information on subjective symptoms and body temperature at the time of onset to admission were obtained from both the health center’s questionnaire and medical interviews after hospitalization. Onset symptoms included body temperature (in some cases only information on febrile or afebrile), cough, runny nose, sore throat, headache, diarrhea, chills, fatigue, loss of taste, loss of smell and body aches or arthralgia. In addition, sputum production, changes in appetite, nasal congestion, chest tightness, constipation, shortness of breath, abdominal pain, nausea or vomiting and insomnia were investigated as pretreatment symptoms. Fever was defined as an axillary temperature of 37.5 degrees Celsius (°C) or higher for those whose temperature was taken, and for those with just febrile or afebrile “febrile” was recorded. The start of the observation period was the date of symptom onset or diagnostic test, whichever came first. The end of the observation period was the date of discharge in patients who did not receive corticosteroid treatment, or the day treatment started.

The present study follows the STROBE (Strengthening the Reporting of Observational Studies in Epidemiology) Statement for observational studies [20], and the check list has been provided in Appendix A.

### 2.2. About the Institution Where the Research Was Conducted

The Kuramochi Clinic Interpark is located in Utsunomiya City, the most populous city in Tochigi Prefecture, Japan. We established RT-PCR system from December 2020, and COVID-19 was enabled to diagnosed in our institution. Of the 31,709 SARS-CoV-2 PCR tests reported to Utsunomiya City from March to May 2021 [21], 2,946 (9.3%) were performed at our institution, and 149 (22.7%) patients who were positive for SARS-CoV-2 PCR test were detected in our institution. In addition, 3,435 suspected or positive COVID-19 outpatients were visited our institution during the observation period, 26 March to 26 May 2021. Our institute was originally for an outpatient, but with the increase in COVID-19 patients, new 10 inpatient beds for COVID-19 were established and started operation on 26 March 2021. Facilities include low-flow oxygen supply, intravenous drug administration, CT, x-ray, simple blood test, and no intensive care unit. As the Japanese Government had taken legal measures to aggressively hospitalize COVID-19 patients, patients diagnosed with COVID-19 were assigned to appropriate hospitals for treatment or isolation at the discretion of the City Public Health Center based on patient backgrounds or subjective symptoms at the time of diagnosis. During the observation period, COVID-19 patients without symptoms of hypoxia were assigned mainly from the Utsunomiya city.

### 2.3. Screening of N501Y Mutation

In our institution, identification of the N501Y was performed by real-time RT-PCR testing using CFX96 Real-Time System (Bio-Rad, Hercules, CA, USA) from saliva samples collected during hospitalization. The point mutation was determined by using fluorescent labeling probes, primer/probe N501Y (RC344A, TaKaRa, Japan). This primer/probe uses the same reaction system as SARS-CoV-2 detection kit (RC300A/RC30JW, TaKaRa, Japan) [22,23], and can be used under the same reaction conditions. If both detection systems showed no response or Ct values > 40, they were assessed to be inconclusive and excluded from the analysis. In the Public Health Institute, identification of the N501Y was performed by real-time RT-PCR under the reaction conditions according to the manual of the National Institute of Infectious Diseases (Japan) [14].

### 2.4. Evaluation of CT Scan

The initial CT scan was taken on admission or at diagnosis. A semiquantitative CT severity score was used to quantify the extent of lung abnormalities [24,25]. Based on the visually involved area, each of the five lung lobes was scored from 0 to 5: 0, no involvement; 1, <5%; 2, 5–25%; 3, 26–49%; 4, 50–75%; and 5, >75% and summed to calculate the total CT score, which ranged between 0 to 25. Blinded to the clinical status of the SARS-CoV-2 strain, two radiologists (K.I. and M.Honda.), who are board-certified members of the Japan Radiological Society with more than 20 years of experience in radiologist, independently evaluated the CTs and scored them. Interobserver reliability was checked by Pearson correlation coefficient (r = 0.935, *p* < 0.001). As the assessment of lung abnormalities was done in more detail, the CT score evaluated by radiologist one (K.I.) was adopted. Clinically and radiological diagnosis of COVID-19 pneumonia was conducted by radiologist one (K.I.) and a respiratory physician (J.K.), who is a board-certified member of the Japanese Respiratory Society. Cases with ground-glass opacities, the crazy-paving pattern and consolidation, which located mainly in the subpleural area and peri-lobular to pan-lobular lesions, were diagnosed as COVID-19 pneumonia. We also reviewed the CT images with CT severity score of 1 or higher given by the radiologist and discussed with the respiratory physician whether they were scars or pneumonia. In all cases, they were in agreement.

### 2.5. Statistics

First, the baseline characteristics of wild-type 501N and N501YV patients were described. Proportions, medians and means were calculated for each item. To evaluate the differences, chi-square for categorical variables, rank sum for skewed continuous variable, or t-test for normal distribution continuous variables was used. A *p*-value < 0.05 was considered statically significant. Second, simple and multivariate logistic regression analysis (adjusted for age and observation period) was performed to assess the differences in the prevalence of clinical symptoms between those infected with wild-type 501N and N501YV for significant variables in the crude models. Third, to investigate the association between N501YV infection and CT severity score, simple and multivariate linear regression analysis were performed to calculate the coefficients and 95% confidence intervals. When to perform CT scan has been reported to be related to the extent of pneumonia [25]. In addition, in the later observation period when N501YV was predominant, older patients could have been preferentially admitted as the number of patients increased. Therefore, the number of days from symptom onset to the date of CT scan and age were adjusted as a mediator. The former variable was set to 0 days in cases with no symptoms at the time of CT scan. Finally, because the marginal interaction between N501YV infection and higher age was observed (*p* = 0.078), univariable regression analysis was performed for the association between age and CT severity score stratified by wild-type 501N and N501YV infection, respectively. All statistical analyses were carried out by Stata version 16.1 (Stata Corp. College Station, TX, USA).

### 2.6. Ethics Statement

Patients included in the study had to give their consent to participate in the study. The written informed consent was obtained from adult participants (≥20 years) or from parents (<20 years) during hospitalization. The samples used for PCR testing to identify N501Y mutation were taken from the saliva of the participants and no additional invasive tests were done for the study. Demographics and clinical findings were referenced from medical records and anonymized prior to analysis. Ethical approval was obtained from Ethics Committee at Tokyo Medical and Dental University (G2021-004).

## 3. Results

### 3.1. Characteristics

During the study period, 64 patients were admitted to the Kuramochi Clinic Interpark, and 62 patients agreed to participate in the study. Adequate test samples were not available from 2 patients. Through screening for N501YV with specific real-time RT-PCR of 60 cases, 8 cases were inconclusive and were excluded. Finally, 24 patients (46.2%) were classified into the N501YV group and 28 (53.9%) into the wild-type 501N group. N501YV first appeared at 15th week of 2021 and became the dominant strain at 18th week of 2021 (Figure 1). The overall number of the patient was highest in 19th week of 2021. The median (interquartile range [IQR]) age was 32.5 (23.5–49) years, mean (± standard deviation [SD]) age was 37.4 ±16.1 and 55.8% was male, with no significant differences between those with N501YV infection and those with wild-type 501N infection. Further, there were no differences in the distribution of age categories, BMI, smoking status, high-risk medical condition for severe COVID-19, birth country, unknown transmission route, COVID-19 clinical grade and laboratory results between the two groups (Table 1). In the investigation of self-reported exposure settings, one pair of wild-type 501N and three pairs of N501YV had contact before admission. The details of the exposure settings were described in Appendix A, and there was no household or healthcare transmission in patients infected with N501YV. There was no difference of mean resting oxygen saturation in the two groups. Nine patients (17.7%; 5 wild-type 501N and 4 N501YV) were completely free of fever or subjective symptoms at the time of COVID-19 diagnostic testing which was conducted for contact tracing. Excluding the above cases, there was no difference in the length of days from onset to admission between the two groups. There were no tertiary care hospital transfers or deaths associated with worsening of the disease.

### 3.2. Clinical Symptoms

Table 2 shows the comparison of subjective symptoms at onset between the groups. The prevalence of body aches or arthralgia was higher in N501YV-infected patients (13.0% vs. 45.0%). On the other hand, there was no significant difference in the prevalence of cough, runny nose, sore throat, headache, diarrhea, chills, fatigue, loss of taste, loss of smell, and febrile.

Pre-treatment symptoms are described in Table 3. The end of the observation period was the date of discharge in patients who did not receive corticosteroid treatment (*n* = 34; 21 in wild-type 501N and 13 in N501YV), or the day treatment started (*n* = 18; 7 in wild-type 501N and 11 in N501YV) (*p* = 0.115). The mea observation period was significantly longer in the N501Y infection group than that of the wild-type 501N infection group (8.4 ± 3.0 days vs. 6.7 ± 2.6 days, *p* = 0.037). Antitussives, inhaled corticosteroids and antipyretics were prescribed depending on their symptoms. Antibiotics were prescribed for patients with possible bacterial infections. The drugs prescribed during the observation period were described in Appendix A and there was no significant difference between the two groups. As antipyretics are available without a prescription, whether it was used or not could not be ascertained from the medical record. The prevalence of fever of 38 °C or higher was significantly higher in N501YV-infected patients than in wild-type 501N-infected patients (32.1% vs. 75.0%). In a multivariate analysis (Table 4), fever over 38 °C remained significantly associated with N501YV (aOR: 6.07, 95%CI: 1. 68 to 21.94).

### 3.3. Radiological Findings

One patient was unable to undergo CT examination due to claustrophobia, but because he had no obvious pneumonia on radiography, he was managed as a mild illness. The median and mean of CT severity score showed no significant difference in the two groups; however, patients infected with N501YV tended to show higher CT score (mean 2.3 ± 2.4 vs. 4.1 ± 4.1, *p* = 0.061). Multivariable regression analysis (Table 5) revealed that the CT score was significantly higher in patients infected with N501YV (β: 1.90, 95%CI: 0.42 to 3.37). Figure 2 shows that the CT severity score was associated with age in both infection groups (β: 0.06, 95%CI: 0.004 to 0.12 and β: 0.15, 95%CI: 0.07 to 0.23). The association between CT score and higher age was marginally modified among patients infected with N501YV (*p* = 0.078, for interaction).

### 3.4. About Patients Who Stayed in Designated Non-Health Facilities before Hospitalization

Appendix A shows the demographics and clinical data of patients who were directly hospitalized and those who stayed in designated non-health facilities before hospitalization. There were five patients who had undergone pre-hospital isolation at designated non-health facilities. All five patients were infected with N501YV, and eventually transferred to the Kuramochi Clinic Interpark Utsunomiya due to a lack of improvement in their symptoms. The mean (±SD) length from onset to hospital admission was 5.8 ± 1.3 days, which was longer than those who were directly hospitalized (*p* = 0.031). Of those five patients, the median (IQR) age was 48 (40–51) years, with a mean (±SD) of 46.2 ± 7.5 years, and the age tended to be higher than that of directly admitted patients. They all had febrile of over 38 °C, and the prevalence of changes in appetite and shortness of breath was more common (12.8% vs. 80.0%, 6.4% vs. 60.0%, respectively) than those who were directly hospitalized. The mean (±SD) CT severity score was 9.0 ± 1.6, which was significantly higher than that of patients who were directly hospitalized (*p* < 0.001); however, there was no significant difference in the mean resting oxygen saturation (*p* = 0.155).

### 3.5. Trends in Age of Patients

The age distribution of the N501YV cases over time showed that in the early weeks of the appearance of the N501YV most of the patients were in their teens and twenties, but in the later weeks the age of the patients increased (Figure 3). During the early stage after the appearance of N501YV (period of 15 to 18th week of 2021), the mean (±SD) age of the patients infected with N501YV was 23.6 ± 5.9 years, significantly younger than that of patients infected wild-type 501N (33.9 ± 12.6 years; *p* = 0.022).

## 4. Discussion

In the present study, the demographics and clinical data in a general practitioner clinic were compared between patients infected with SARS-CoV-2 wild-type 501N and those with N501YV. There was no significant difference in demographics; however, fever of over 38 °C during the observation period was more common in those infected with N501YV. In multivariate analysis adjusted for patient age and observation period, patients with N501YV showed significantly higher odds of fever of 38 °C or higher than those with wild-type (aOR: 6.07, 95%CI: 1.68 to 21.94). Prior literature also reported that fever was significantly associated with the N501Y mutant strain in France and the UK [10,11]. We add to the literature that fever of 38 °C or higher was found to be more frequent in those with the N501Y mutant strain. In the follow-up of four N501YV-infected patients who were asymptomatic at the time of diagnostic testing, two patients had a fever of over 38 °C within 0 or 1 day. Although the number of cases is small, it is possible that N501YV-infected patients had characteristics that led to more apparent symptoms.

During the study period, there were 536 (103 cases per 100,000 people) newly diagnosed COVID-19 cases in Utsunomiya City [26]. In situations where inpatient beds were unavailable with the rapid increase in COVID-19, patients were sometimes assigned to isolation in designated non-health facilities. Four of the five patients who had undergone pre-hospital isolation at designated non-health facilities were diagnosed at weeks 19 to 20th of 2021, when it was the peak of the fourth wave of the COVID-19 pandemic in Japan. The patients did not experience a decrease in oxygen saturation; however, they tended to be older and had severe pneumonia than those who were admitted to a hospital directly, suggesting that oxygen saturation alone is insufficient for monitoring and that evaluation of pneumonia by CT scan is necessary, especially if a post-middle-aged patient is to be treated in a designated non-health facility rather than in a hospital.

N501YV infection was also associated with an increase in CT score. This is consistent with the findings from a study in China comparing radiological imaging of B.1.1.7-infected patients with non-VOC infected patients. The study also suggested that B.1.1.7 infection may cause more severe pneumonia [27]. Therefore, finding of the present study is consistent with the idea that an increased risk of ICU admission and mortality in N501YV-infected patients [7,8] is due to the differences in the severity of pneumonia, which is dependent on the SARS-CoV-2 strain.

During the early stage of the appearance of the N501YV (period of 15 to 18th week of 2021), the mean age of the patients infected with the N501YV was significantly younger than that of patients infected with wild-type 501N, suggesting that this variant first spread to younger age groups followed by middle-aged and older age groups. Similarly, in the UK, the difference in the proportion of SGTF and non-SGTF cases among secondary school-aged children (11–18 years) was temporarily statistically significant in the early stages of the SGTF outbreak [28]. In France, the B.1.1.7 variant was also estimated to have spread to young people (<20) before to older age people [29]. This trend may be attributed to continuing school activities even during the lockdown, or to inadequate protection against infection in younger age groups. At period between Weeks 15 to 18 of 2021, of the four cases with known transmission route that were infected N501YV, two were infected at the workplace while two were infected at social gatherings. Since the transmission route of the other seven could not be traced, it is difficult to determine from this study the factors responsible for this epidemic pattern in Japan.

There are several limitations in this study. First, sample size was small due to a single-clinic study (52 cases). However, inpatient managements were conducted by two full-time clinicians, and the interview at the time of admission or during the course of hospitalization was standardized, which could reduce bias among examining physicians compared to larger facilities. Second, because the pressure on the healthcare delivery system was greater in the period when N501YV was dominant, hospitalized patients may be in a more serious condition due to more stringent selection for hospitalization (selection bias), or because some patients had to go through a longer wait to be admitted. At the time when the City Public Health Center assigned the patients, pneumonia was undetermined in all but one case. Therefore, the main criteria for admission were older age, high-risk medical conditions and severe onset symptoms. Indeed, the mean age of patients diagnosed in the second half of the study week was 10 years older than that in the first half (mean [±SD] 32.7 ± 14.2 years old vs. 42.1 ± 16.9 years old, *p* = 0.036). However, the prevalence of medical risk and fever at onset was not significant between the two study periods. It is noteworthy that all but one patient admitted to the Kuramochi Clinic Interpark did not require oxygenation and were not admitted for respiratory failure throughout the observation period. Therefore, the potential confounder of local hospital pressure was accounted for indirectly through adjusting for age and the number of days from symptom onset to CT scan or observation period. Third, the detection of the N501Y mutation was done by specific real-time RT-PCR testing and not by whole genome analysis. Hence, cases with mutations in other genomic regions may have been included in the N501Y group. However, whole genome analysis of all patients is not clinically feasible in terms of time and cost. As mentioned, the prevalence of VOCs in Japan during the study period indicates that N501YV-PCR positive strain detected in this study could be thought to be a proxy for the B.1.1.7 lineage. Fourth, the history of antipyretics use was unclear as some patients had used over-the-counter medications prior to admission. The use of antipyretics may have affected the fever level or symptoms. However, over-the-counter antipyretics were available to anyone, anywhere throughout the observation period. Therefore, we think that this is not an issue when comparing the prevalence of fever or symptoms.

Nonetheless, present study added to the literature that patients with N501YV showed higher prevalence of high fever and extensive pneumonia than patents with wild-type 501N, which may explain the high ICU admission rate and mortality among patient with N501YV [7,8]. Although the number of whole genome analyses and specific PCR tests is limited, it was suggested that careful observation of the clinical characteristics may be able to suspect changes in the predominant SARS-CoV-2 strains in the area.

## 5. Conclusions

In this study, it was revealed that N501YV-infected patients had a higher tendency to experience high fever and severe pneumonia compared with wild-type 501N-infected patients in a general practice setting. There is a need to continue to investigate the differences in clinical symptoms among emerging SARS-CoV-2 strains.

## Figures and Tables

**Figure 1 jcm-10-05865-f001:**
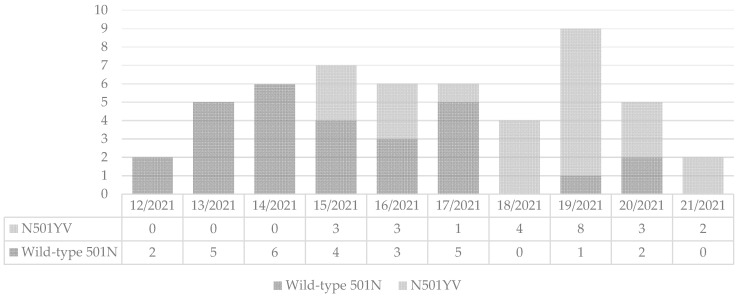
Changes in the number of N501YV infection diagnosis per week.

**Figure 2 jcm-10-05865-f002:**
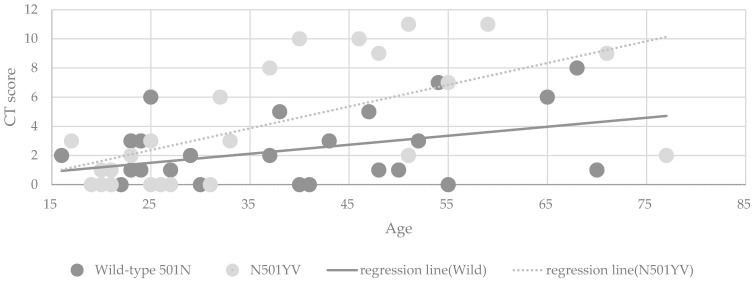
CT severity score distribution of COVID-19 patients per age. (*n* = 51).

**Figure 3 jcm-10-05865-f003:**
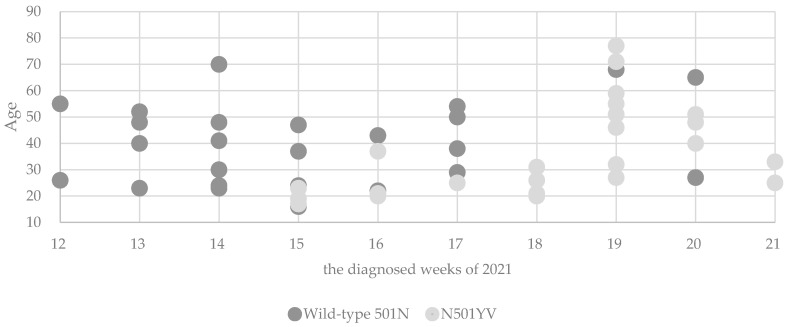
Age distribution of COVID-19 patients per week.

**Table 1 jcm-10-05865-t001:** Baseline characteristics for the patients infected with wild-type 501N and N501YV.

	Total		Wild-Type 501N		N501YV		
	*n* = 52		*n* = 28		*n* = 24		*p*-Value
Median age (years)	32.5	IQR 23.5–49	37.5	IQR 24–49	31.5	IQR 22–49.5	0.545
Average age (years)	37.4	±16.1	38.2	±15.5	36.5	±17.1	0.700
Age category							
Less than 19 years, *n* (%)	3	(5.8)	1	(3.6)	2	(8.3)	0.850
20–39 years, *n* (%)	27	(52.0)	14	(50.0)	13	(54.2)	
40–59 years, *n* (%)	17	(32.7)	10	(35.7)	7	(29.2)	
Over 60 years, *n* (%)	5	(9.6)	3	(10.7)	2	(8.3)	
Sex							
Male, *n* (%)	29	(55.8)	15	(53.6)	14	(58.3)	0.964
Female, *n* (%)	23	(44.2)	13	(46.4)	10	(45.5)	
BMI							
Under weight (<18.5), *n* (%)	5	(9.6)	4	(17.3)	1	(4.2)	0.519
Normal weight (18.5–23.9), *n* (%)	25	(48.1)	11	(39.3)	14	(58.3)	
Overweight (24.0–27.9), *n* (%)	13	(25.0)	7	(25.0)	6	(25.0)	
Obesity (28+), *n* (%)	7	(13.5)	5	(17.9)	2	(8.3)	
Missing, *n* (%)	2	(3.9)	1	(3.6)	1	(4.2)	
Smoking status (current)							
No, *n* (%)	38	(73.1)	23	(82.1)	15	(73.1)	0.256
Yes, *n* (%)	12	(23.1)	4	(14.3)	8	(23.1)	
Missing, *n* (%)	2	(3.9)	1	(3.6)	1	(3.9)	
High-risk medical condition for severe COVID-19							
Yes, *n* (%)	11	(21.2)	8	(28.6)	3	(12.5)	0.157
Birth country							
Japan, *n* (%)	37	(71.2)	19	(67.9)	18	(75.0)	0.571
Other country, *n* (%)	15	(28.9)	9	(32.1)	6	(25.0)	
Unknown transmission route							
Yes, *n* (%)	26	(50.0)	12	(42.9)	14	(58.3)	0.266
Clinical grade based on the NIH guidelines							
Asymptomatic, *n* (%)	2	(3.9)	0	(0)	2	(8.3)	0.143
Mild illness, *n* (%)	24	(46.2)	16	(57.1)	8	(33.3)	
Moderate illness, *n* (%)	25	(48.1)	12	(42.9)	13	(54.2)	
Severe illness, *n* (%)	1	(2.0)	0	(0)	1	(4.2)	
Laboratory results							
White blood cell (/μL)	4787	±1610	4607	±1391	4996	±1841	0.391
Hemoglobin (g/dL)	14.5	±1.4	14.4	±1.5	14.7	±1.3	0.347
Platelet (* 10,000/μL)	21.6	±5.6	22.1	±5.1	20.9	±6.3	0.448
Alanine transaminase (U/L)	24	±13	22	±12	26	±15	0.399
Creatinine (mg/dL)	0.81	±0.17	0.80	±0.16	0.82	±0.19	0.653
Albumin (g/dL)	4.5	±0.3	4.5	±0.3	4.5	±0.3	0.826
D-dimer (μg/mL)	0.56	±0.42	0.53	±0.44	0.60	±0.41	0.608
Average resting oxygen saturation	98.8	±1.0	99.0	±0.6	98.6	±1.3	0.192
Stay in designated non-health facilities before hospitalization							
Yes, *n* (%)	5	(9.6)	0	(0)	5	(20.8)	**0.011**
	Total *		wild-type 501N		N501YV		
	*n* = 43		*n* = 23		*n* = 20		
Average period of onset to admission (days)	3.7	±2.3	3.6	±2.0	4.0	±2.6	0.587
	Total **		wild-type 501N		N501YV		
	*n* = 51		*n* = 27		*n* = 24		
Average CT severity score	3.1	±3.4	2.3	±2.4	4.1	±4.1	0.061

Boldface indicates statistical significance (*p* < 0.05). Abbreviations: BMI, body mass index; the NIH, the National Institute of Health. High-risk medical condition for severe COVID-19: type 2 diabetes mellitus, hypertension, cardiovascular disease, chronic obstructive pulmonary disease, chronic kidney disease, cancer, obesity and dyslipidemia. * Nine patients who were asymptomatic at the time of COVID-19 diagnostic test which was conducted for contact tracing were excluded. ** One patient who did not undergo CT scan was excluded.

**Table 2 jcm-10-05865-t002:** Onset symptoms of the patients infected with wild-type 501N and N501YV. (*n* = 43 *).

	Total		Wild-Type 501N		N501YV		
	*n* = 43		*n* = 23		*n* = 20		*p*-Value
Onset symptoms							
Cough, *n* (%)	19	(44.2)	9	(39.1)	10	(50.0)	0.474
Runny nose, *n* (%)	9	(20.9)	7	(30.4)	2	(10.0)	0.100
Sore throat, *n* (%)	10	(23.3)	6	(26.1)	4	(20.0)	0.637
Headache, *n* (%)	17	(39.5)	10	(43.5)	7	(35.0)	0.571
Diarrhea, *n* (%)	2	(4.7)	2	(8.7)	0	(0)	0.177
Chills, *n* (%)	6	(14.0)	1	(4.4)	5	(25.0)	0.051
Fatigue, *n* (%)	19	(44.2)	9	(39.1)	10	(50.0)	0.474
Loss of taste, *n* (%)	0	(0)	0	(0)	0	(0)	-
Loss of smell, *n* (%)	0	(0)	0	(0)	0	(0)	-
Body aches or arthralgia, *n* (%)	12	(27.9)	3	(13.0)	9	(45.0)	**0.020**
Onset fever							
Afebrile, *n* (%)	15	(34.9)	10	(43.5)	5	(25.0)	0.205
Febrile, *n* (%)	28	(65.1)	13	(56.5)	15	(75.0)	

Boldface indicates statistical significance (*p* < 0.05). * Nine patients who were asymptomatic at the time of COVID-19 diagnostic test which was conducted for contact tracing were excluded.

**Table 3 jcm-10-05865-t003:** Pre-treatment symptoms of the patients infected with wild-type 501N and N501YV. (*n* = 52).

	Total		Wild-Type 501N		N501YV		
	*n* = 52		*n* = 28		*n* = 24		*p*-Value
Average observation period (days)	7.6	±2.9	8.4	±3.0	6.7	±2.6	**0.037**
Pretreatment symptoms							
Cough, *n* (%)	35	(67.3)	18	(64.3)	17	(70.8)	0.616
Runny nose, *n* (%)	17	(32.7)	11	(39.3)	6	(25.0)	0.274
Sore throat, *n* (%)	21	(40.4)	13	(46.4)	8	(33.3)	0.337
Headache, *n* (%)	31	(59.6)	15	(53.6)	16	(66.7)	0.337
Diarrhea, *n* (%)	8	(15.4)	6	(21.4)	2	(8.3)	0.192
Chills, *n* (%)	15	(28.9)	5	(17.9)	10	(41.7)	0.059
Fatigue, *n* (%)	37	(71.2)	17	(60.7)	20	(83.3)	0.073
Loss of taste, *n* (%)	10	(19.2)	7	(25.0)	3	(12.5)	0.254
Loss of smell, *n* (%)	9	(17.3)	6	(21.4)	3	(12.5)	0.396
Body aches or arthralgia, *n* (%)	14	(26.9)	5	(17.9)	9	(37.5)	0.111
Sputum production, *n* (%)	13	(25.0)	7	(25.0)	6	(25.0)	1.000
Changes in appetite, *n* (%)	10	(19.2)	5	(17.9)	5	(20.8)	0.786
Nasal congestion, *n* (%)	10	(19.2)	6	(21.4)	4	(16.7)	0.664
Chest tightness, *n* (%)	7	(13.5)	4	(14.3)	3	(12.5)	0.851
Constipation, *n* (%)	2	(3.9)	2	(7.1)	0	(0)	0.182
Shortness of breath, *n* (%)	6	(11.5)	2	(7.1)	4	(16.7)	0.284
Abdominal pain, *n* (%)	5	(9.6)	2	(7.1)	3	(12.5)	0.514
Nausea or vomit, *n* (%)	3	(5.6)	2	(7.1)	1	(4.2)	0.646
Insomnia, *n* (%)	5	(9.6)	4	(14.3)	1	(4.2)	0.217
Fever							
Febrile over 37.5 °C, *n* (%)	39	(75.0)	18	(64.3)	21	(87.5)	0.059
Febrile over 38 °C, *n* (%)	27	(51.9)	9	(32.1)	18	(75.0)	**0.001**

Boldface indicates statistical significance (*p* < 0.05).

**Table 4 jcm-10-05865-t004:** The associations of pre-treatment symptoms between wild-type 501N and N501YV infection, using univariable and multivariable logistic regression models. (*n* = 52).

	Febrile Over 38 °C				
		Crude		Adjust	
	*n*	OR	95%CI	OR	95%CI
Total, *n* = 52					
Wild-type 501N	9	Ref.		Ref.	
N501YV	18	6.33	1.87 to 21.40	6.07	1.68 to 21.94

Adjust: age (continuous) and observation period.

**Table 5 jcm-10-05865-t005:** The associations of CT score with N501YV infection, using univariable and multivariable linear regression models. (*n* = 51 **).

Variables	Crude		Adjust	
	β	95%CI	β	95%CI
Wild-type 501N	Ref.		Ref.	
N501Y	1.78	−0.09 to 3.66	1.90	0.42 to 3.37
Age	-	-	0.12	0.07 to 0.16
Number of days from symptom onset to CT scan	-	-	0.49	0.18 to 0.80

Adjust: age, number of days from symptom onset to CT scan. ** One patient who did not undergo CT scan was excluded.

## Data Availability

The data are available on reasonable request from the corresponding author. For privacy restrictions, the data are not publicly available.

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
