# Peer review of "Clinical Characteristics of Patients with SARS-CoV-2 N501Y Variants in General Practitioner Clinic in Japan"

_jcm, 2021, doi:10.3390/jcm10245865_

Round 1
Reviewer 1 Report
Dear Author,
Thank you, the paper is well written and figures and tables are good
Some corrections needed:
Introduction: should be shortened. It is too long
ethics statement: a reference number for the ethical commitee
Results:
My major concern is why there is no results of laboratory results? You compared CT scan but not laboratory results?
Several laboratory markers may be interesting, in particular ferritin, neutrophil to lymphocyte ratio, ddimer which is known to be associated with poor outcome (please read and cite this article: Yildiz H, et al Validation of Neutrophil-to-Lymphocyte Ratio Cut-off Value Associated with High In-Hospital Mortality in COVID-19 Patients. Int J Gen Med. 2021 Sep 1;14:5111-5117. doi: 10.2147/IJGM.S326666. PMID: 34511993; PMCID: PMC8420786.)
Reviewer 2 Report
- Does this study follow STROBE Statement? If so, it is recommended to add this section in the methods section and provide a checklist.
- How was the sample size of this study determined?
- It is suggested to add a part of content in the section of methods and materials to explain the basic information of the hospital where the study is conducted, such as the size of the hospital, diagnosis and treatment level, main sources of patients and so on.
- It is recommended to supplement the diagnostic criteria for COVID-19 cases in section 2.1. Andto add relevant references to the diagnostic criteria of genotype screening in Section 2.2.
- Line 121: Is fever defined as armpit temperature or oral temperature?
- Among enrolled patients, the end of the observation period was the date of discharge or treatment initiation for patients who did not receive corticosteroid therapy. It is recommended that patients who do not receive corticosteroids be given a detailed description of the treatment taken and whether it affects the outcome measures.
- The clinical and radiological diagnosis of COVID-19 pneumonia was carried out by a specialist and a respiratory physician. It is recommended to add descriptions of the consistency and accuracy of the two judgements, as well as how to deal with differences.
- It is suggested that the description and analysis of the patients' pre-admission epidemiological history should be added to the results to ensure that these patients are not from concentrated communities or known neighbors, etc. Because the sample size of this study is rarely, if included in patients from the close to the area, for example a genotype infection in an elderly community, under the limited sample size, can be concluded that the relationship between age and virus genetic, but the result is likely to be only occasionally, the research conclusion is not true.
- What is the standard of "Risk of severe COVID-19" in Table 1and the difference between "severe illness" in "Severity Based on the NIH guidelines"? Is there any overlap between the two categories?
- Is it more appropriate to select one of the results of "Median CT severity score" and "Average CT severity score" in Table 1, or what is mainly expressed by selecting both results to present.
- It is suggested to discussthe benefits that will be provided for future diagnosis and treatment as well as clinical practice and prognosis of the patients by observing the clinical and imaging features of N501Y variants.
Round 2
Reviewer 1 Report
/
Reviewer 2 Report
We see in the current article version for all the questions have the corresponding added and modified, although still exist in the aspect of design and analysis methods of the study can be improved, we still think first of all, this study is the theme of the novel, the general situation is relatively complete, is a worthy of reading research.